# Do Primary Health Professionals in Brazil Have a Misperception about Food? The Role of Food Literacy as a Positive Predictor

**DOI:** 10.3390/nu16020302

**Published:** 2024-01-19

**Authors:** Larissa Baungartner Zeminian, Ligiana Pires Corona, Marcela Chagas da Silva, Isabelle do Nascimento Batista, Diogo Thimoteo da Cunha

**Affiliations:** 1Laboratório Multidisciplinar em Alimentos e Saúde, Faculdade de Ciências Aplicadas, Universidade Estadual de Campinas—UNICAMP, Rua Pedro Zaccaria n° 1300, Limeira 13484-350, SP, Brazil; larissab.nutricionista@gmail.com (L.B.Z.); m202555@dac.unicamp.br (M.C.d.S.); i237098@dac.unicamp.br (I.d.N.B.); 2Laboratório de Epidemiologia Nutricional, Faculdade de Ciências Aplicadas, Universidade Estadual de Campinas—UNICAMP, Rua Pedro Zaccaria n° 1300, Limeira 13484-350, SP, Brazil; ligiana.corona@fca.unicamp.br

**Keywords:** diet, food and nutrition, risk perception, food preferences, knowledge

## Abstract

Risk perception is socially constructed; psychological elements control people’s reactions to a hazard, and even health professionals may have difficulty determining what healthy food is. This work aimed to measure food literacy and food risk perceptions among primary healthcare professionals in a Brazilian city. In the first phase, 280 health professionals working in primary care in Rio Claro, Brazil, were studied. The Short Food Literacy Questionnaire (SFLQ-Br) and scales of risk and benefit perception of 50 foods were used. In the second phase, 20 professionals were interviewed to investigate the responses to different foods observed in the first phase. In this second phase, 16 users of the health system were also enrolled to understand their perceptions and how the nutrition messages conveyed by the health team reached them. Professionals scored an average of 34.5 on food literacy (for which there is a maximum score of 52). They showed difficulty with dietary guidelines and their interpretation. Food’s risk and benefit perception were generally consistent with the recommendations of the Food Guide for the Brazilian Population. However, some processed foods or those with no proven health benefits were considered healthy by the study participants, indicating a biased perception (e.g., gelatin, processed turkey breast, cream crackers, and cereal bars). Less misperception was observed when food literacy was higher, which positively predicted risk perception. The reasons for identifying benefits of these foods ranged from the false impression that they are natural and nutritious foods to the comparative claim that they are better for health than similar foods. The results indicate the need to educate health professionals based on current references to avoid bias in population counseling.

## 1. Introduction

Food choices are influenced by several factors, such as biological (hunger and appetite), psychological (mood, stress, and guilt), physical (access and cooking skills), social (culture and family), and economic (cost, income, and availability) factors [1,2]. Brazilian food consumption varies by age group, with the consumption of ultra-processed foods tending to decrease with age; by urbanization, as people living in rural areas consume staple foods more frequently and have better diet quality, while people living in urban areas eat more meals outside from home [3]; and by income, with the worst-quality diets being consumed by lower-income people [4]. However, the general population’s perceptions of food may be related to their behavior and lifestyle patterns. For example, normal-weight and active individuals classify their diets as balanced, varied, and complete when fruits and vegetables are included. On the other hand, people with chronic, non-communicable diseases tend to associate a healthy diet with avoiding fatty and sweet products [5]. Environmental, political, and health motivations [5]; age; beliefs; and knowledge about food [1] also influence food choices. Knowledge is understood as a prerequisite for change toward health promotion through cognitive means. Lack of knowledge, in turn, results in individuals having little reason to change their behavior [6]. Although essential, it is known that the relationship between knowledge and practice can be complex, given the different perceptions and pressures associated with practice [7].

Food literacy (FL) can be understood as a factor that includes knowledge about nutrition, the ability to communicate about food issues, critical reflection on behavior, food consumption, and practical skills in food planning, selection, and preparation [8]. Some authors also define FL as a factor that builds autonomy, confidence, and problem-solving skills (see Silva et al. (2023) [9] for more definitions). FL is considered necessary for public health because it represents a promising approach to helping to solve problems ranging from obesity to environmental sustainability [10]. FL extends the scope health literacy [9], which can be understood as the set of skills required for a healthy lifestyle [10]. The cognitive and social skills that motivate individuals to lead healthy lifestyles ratify FL in ways that can be categorized as functional (an individual’s ability to find and understand health-related information), interactive (the ability to receive and share information about health in one’s environment), and critical (the ability to critically evaluate and question health information) [8]. FL has been shown to have positive repercussions via its role as an essential influence on eating behavior. FL builds resilience and independence by including information on eating skills (techniques, knowledge, and planning skills), confidence to improvise and problem solve, and the ability to access and share information [10]. According to Morgan et al. (2023) [11], experiences that improve FL can improve food quality and reduce food insecurity. Improving the population’s FL through health communication, whether in care, health services, or mainstream media, is necessary. In this sense, to promote public health and food communication, health professionals must be equipped with adequate FL, knowledge, and social marketing skills [9]. Limited FL is a barrier for adequate practical advice since FL can empower communities, dietary resilience, and food sovereignty [12].

In Brazilian health policy, many strategies are being considered to ensure the constitutional right to health [13]. Brazil is an example of a country improving the health of its population through access to primary health care [14] because primary health care is developed through integrated care practices and qualified management, carried out with a multi-professional team, targeting the population in a specific geographic area for which the teams assume health responsibility. Primary health care in Brazil takes place in the basic units and, if necessary, at home and in other community spaces. It is aimed at individuals and their families at all stages of life. Health promotion, protection, and recovery actions are offered through the welcoming of users; individual care (consultation) provided by a physician, nurse, or dentist; collective care (educational groups); visits; and home care [15]. Despite involving several professionals, this primary health care is primarily performed by physicians and nursing professionals, and a nutritionist is not listed as a mandatory member of these teams [15]. The quality and effectiveness of a primary health care service is determined by the professionalism of its staff [16], showing the importance of health professionals at this level of care and the need to invest in their training [17]. The problem is that many professionals do not have the skills and appropriate tools with which to relate individual care to the population or community [18].

In nutrition, primary healthcare professionals face the challenge of providing food and nutrition knowledge to individuals. The lack of awareness among primary healthcare professionals, potentiated by inadequate nutrition knowledge, can lead to negative beliefs about the effectiveness of nutrition interventions. It may also lead to nutrition education interventions based on personal experience rather than scientific evidence [19]. Brazilian primary healthcare professionals may have less knowledge and low self-efficacy perceptions of using the Food Guide for the Brazilian Population than nutritionists/dietitians. The Food Guide is a Ministry of Health document, a reference for food and nutrition interventions in primary health care [20]. The guide’s new edition has undergone significant restructuring, taking the focus away from nutrient groups and shifting to the NOVA classification, which professionals may not understand well [21]. In Brazil, there is a widespread perception that food and nutrition education is the exclusive responsibility of dietitians, resulting in fragmented interventions that do not meet the population’s needs [22,23]. However, nutrition education is a task shared by other healthcare team members.

This study proposes the following hypotheses:

(a) The FL of health professionals corresponds to medium to low levels;

(b) Many processed and ultra-processed foods are perceived as being healthy by health professionals;

(c) FL can be a positive predictor of risk perception regarding processed and ultra-processed foods;

(d) Many healthy attributes of foods are based on common sense rather than scientific evidence.

With this in mind, this study aimed to measure FL and risk and benefit perceptions regarding food among primary healthcare professionals in Brazil.

## 2. Methods

We used a mixed-methods approach for this study. The first step was the collection and analysis of quantitative data. The second step was the collection and analysis of qualitative data. The second step was planned and developed based on the results of the first step, constituting a sequential explanatory design [24].

### 2.1. First Step—Quantitative Approach

#### 2.1.1. Sample

All primary health professionals from Rio Claro, Brazil (n = 327), were invited to participate in this study. The acceptance rate was 85.6%; i.e., 280 health professionals participated. In Brazilian primary care, there are professionals with higher education (physicians, nurses, and dentists), technical-level education (nursing technicians), and high-school-level education (community health agents). A community health agent is responsible for monitoring, through home visits, all families and individuals for whom they are responsible and for developing activities to allow the population to work with the health team to promote health promotion, protection, and prevention [15]. In this study, these agents were professionals of both genders. Participation was voluntary, and all participants gave free and informed consent. The study methods were approved by the Research Ethics Committee of the *Universidade Estadual de Campinas* (CAAE: 46398221.4.0000.5404).

#### 2.1.2. Measures

Two questionnaires were used to collect quantitative data, namely, the “Short Food Literacy Questionnaire validated for Brazil” (SFLQ-Br) and the “risk and benefit perceptions about food”, both conducted in person using tablets.

The SFLQ is an appropriate tool with which to assess food literacy [10]. We used the version validated for Brazilian adults, the SFLQ-Br [25]. It consists of an objective questionnaire with 12 indicators. Responses are given on a four- or five-point scale, e.g., 1—totally disagree to 5—totally agree; 1—very poor to 4—very good; 1—very difficult to 5—very easy; or 1—never to 5—always. A food literacy score was calculated using the sum of each item [10].

The risk and benefit perceptions about food questionnaire allows for the identification of the public perceptions of 50 food items [26]. Based on the Food Guide for the Brazilian Population [21], the questionnaire assesses how foods affect health. Participants rated each food on a 7-point scale ranging from −3 (it is very bad for one’s health) to +3 (it is very good for one’s health). Positive means refer to “Perceived healthy foods”, while negative means refer to the classification “Perceived unhealthy foods”. For foods classified as “not bad for health or good for health”, a mean close to zero was assumed.

#### 2.1.3. Data Analysis

For data analysis, the theoretical distributions of the variables were first analyzed using means, variances, skewness, kurtosis, and the histogram of the distribution. The Kolmogorov–Smirnov test (with Lilliefors correction) was used to check the normality of the data. The SFLQ-Br and the food risk and benefit perception questionnaire were subjected to confirmatory factor analysis (CFA). This CFA was conducted to ensure the quality of the factors of each questionnaire, as both questionnaires were used with healthcare professionals for the first time. The CFA followed the original structure of the two questionnaires (Zeminian et al., 2022 [22] and Marsola et al., 2021 [23]). The SFLQ-Br has one original factor, and the food risk and benefit questionnaire has seven factors: healthy stereotyped food, cafeteria, typical Brazilian staple foods, fad diet food, ultra-processed foods, foods for special dietary uses, and oils and fat. CFA was conducted using diagonally weighted least squares (DWLS). Each indicator should have a factor loading higher than 0.35. The chi-squared value (χ^2^ with *p* < 0.05), the comparative fit index (CFI > 0.90), the root mean square error of approximation (RMSEA < 0.10), the Tucker–Lewis index (TLI > 0.90), the standardized root mean square residual (SRMR < 0.10), and the goodness-of-fit index (GFI > 0.90) were used to test model fit [27]. Composite reliability (CR) was used to assess the reliability of the factors.

Seven multiple linear regression models were constructed, i.e., one for each food risk and benefit perception factor. The models included the food literacy score (continuous) as an independent variable. All models were adjusted for gender (binary), age (discrete), and previous participation in any nutritional education course (binary). The goodness of fit of each model was assessed using histograms of the residuals, Q.Q. plot analysis, and the Cramér–von Mises test (*p* > 0.05). Collinearity was measured by checking each variable’s variance inflation factor value (<3.3).

Analyses were performed using JASP 0.17.2.1 (JASP Team, University of Amsterdam, Amsterdam, The Netherlands) and SmartPLS 4 (SmartPLS GmbH., Boenningstedt, Germany).

### 2.2. Second Step—Qualitative Approach

#### 2.2.1. Sample

The qualitative approach had two target groups: (i) primary healthcare professionals who had participated in the quantitative phase and (ii) users of the primary healthcare care program in Rio Claro, Brazil. The users were included as a control and to see how the food and nutrition messages reached them.

Four health professionals from five different health units were invited to be part of the first target group. The units were selected to cover different city regions. One participant from each professional category per unit was invited: physician, nurse, community health agent, and dentist. The invitation was randomly distributed among the five different units. One professional declined the invitation, whereupon a second professional was invited. In this way, 20 professionals participating in the previous quantitative phase were selected. Users were approached while waiting for healthcare in the waiting room at the health unit. Sixteen people of both genders who had given informed consent and were over 18 participated in the study. People with any level of intellectual disability were not invited.

#### 2.2.2. Interview and Analysis

Face-to-face interviews were used. The face-to-face interview is the most commonly used strategy in fieldwork [28]. It is a conversation conducted on the interviewer’s initiative to gather information relevant to the research subject and is considered a privileged form of social interaction [28]. The qualitative approach aimed to explore two defined research questions: (a) what are the reasons for health professionals’ perceptions of food, particularly healthy stereotyped food, and (b) how do users of healthy units perceive healthy stereotyped food?

The interviews were structured as and characterized by a combination of questions requiring the interviewees to express their opinions on the topic, with information derived from their reflection on reality and their experiences [28]. Both target groups were asked questions about why they thought certain foods are healthy or unhealthy; e.g., “Why do you think “corn chips” are unhealthy? The foods were strategically selected after analyzing the quantitative data. Specifically, for the unhealthy food question, foods with a rating of −1.0 or less were selected; for the neutral food question, foods with a rating between −0.9 and 0.9 were selected; and for the healthy food question, four foods were selected from the stereotyped healthy food.

All the content of the in-depth interviews was transcribed and analyzed using Laurence Bardin’s qualitative thematic content analysis method [29]. This method divides speeches into meaning cores from which non-prioristic categories emerge. The principal researcher determined the categories, which were later reviewed independently by another researcher for validation and grouping. A consensus was reached via a final discussion. Qualitative data were analyzed using MAXQDA© 2022 software—VERBI GmbH 2018 (Berlin, Germany).

## 3. Results

### 3.1. Quantitative Approach

A sample of 280 health professionals from all the health units of the city (n = 17) participated in the quantitative step. The characteristics of the participants are described in Table 1. The number of women was significantly higher than that of men (86.43%). 

Almost half (49.29%) of the participants were community health agents. More than half of the participants (58.93%) reported that they had never taken part in a food and nutrition course.

Table 2 shows the results of using the SFLQ-Br with health professionals. All indicators showed adequate factor loadings (>0.40). The SFLQ-Br construct showed adequate composite reliability (CR = 0.82). The professionals scored an average of 34.5 on food literacy (for which there is a maximum score of 52). The professionals indicated difficulty understanding and applying the concepts of the Brazilian population dietary guidelines (indicators 3, 4, and 5). There was a significant difference between professionals with complete higher education (mean = 35.8; 6.8) and other professionals (mean = 33.2; 6.5; t = 3.23; *p* = 0.001).

The CFA of the risk and benefit perception of food exhibited an adequate fit: χ^2^ = 1733.32 (*p* < 0.001); RMSEA = 0.09; SRMR = 0.09; CFI = 0.93; TLI = 0.92; and GFI = 0.95. The professionals’ perceptions of food health risks and benefits were determined using seven factors grouped according to the CFA (Table 3). Factor 1 includes several ultra-processed foods classified as healthy or neutral (neither good nor bad for one’s health). This classification highlights foods that are stereotypically considered healthy. Factors 2 and 5, on the other hand, correspond to ultra-processed foods that are classified as harmful to one’s health. Foods associated with fad diets (factor 4) and typical Brazilian staples (factor 3) were generally classified as healthy. Foods grouped in factors 6 and 7 were classified as neutral.

Regarding the regression models, the food literacy scores were associated with some factors concerning food risk and benefit perceptions. Food literacy scores showed negative coefficients for factor 1—health stereotyped food (β = −0.12; *p* = 0.04), factor 2—cafeteria (β = −0.22; *p* < 0.001), factor 5—ultra-processed food (β = −0.14; *p* = 0.02), and factor 7—oils and fat (β = −0.13; *p* = 0.03). In this case, food literacy was a positive predictor; i.e., the higher the food literacy, the lower the perception of the healthiness of these foods. On the other hand, food literacy scores were not significant for factor 3 (β = 0.07; *p* = 0.25), factor 4 (β = 0.10; *p* = 0.08), and factor 6 (β = 0.04; *p* = 0.45). All models showed residuals without bias and the absence of collinearity.

### 3.2. Qualitative Approach

Of the 20 health professionals interviewed, 5 were physicians, 5 were nurse assistants, 5 were community health agents, and 5 were dentists. The age range was from 20 to 59 years. The majority (70%) were women, half had higher education, and 35% had completed high school. The majority (70%) reported that they had never taken part in a course on food and nutrition.

All users (n = 16) were adults (aged 18 to 59 years), and the vast majority (87.5%) were women. In terms of education, the majority had completed secondary education (31%), but a significant number had not completed primary education (25%). Half of the respondents were formally employed, but a considerable proportion (37.5%) was unemployed. Regarding the family income of these participants, 12.5% received less than one minimum wage per month (<USD 263.00), half received between one and two minimum wages, 19% received between two and three minimum wages, and 18.5% received above three minimum wages. Finally, regarding the number of people per household, most participants (69%) reported living in a home with between three and four people.

Based on the quantitative step, some foods were selected for interviews to explore perceptions of the risks and benefits of these foods and their effects on health. These were corn chips, stuffed cookies, and fruit in syrup (with negative mean scores, suggesting they are harmful to health); cream crackers, ham, soy oil, and light foods (with positive or negative mean scores near zero, suggesting they are neither good nor bad for health); and processed turkey breast, gelatin, coconut oil, and honey (with positive mean scores, suggesting they are good for health).

Figure 1 shows two main categories for foods perceived as unhealthy: poor in nutrients and degree of processing. The professionals argued that these foods have excess sugar, fat, or salt; are low in nutrients; and contain artificial dyes and preservatives.

Figure 2 shows the categories and subcategories that emerged for foods considered neither good nor bad for health. Professionals showed ambivalent attitudes toward these foods, pointing to positive and negative attributes. The negative categories and subcategories were similar to those for foods perceived as unhealthy. However, professionals could highlight good characteristics of these foods, such as their inexpensiveness or having a better composition than a similar food. For these foods, there was a neutral category in which the professionals weighed the amount consumed.

Finally, Figure 3 shows the categories and subcategories that emerged for the perceived healthy foods. As with the foods in the previous question, the perceived healthy foods also provoked ambivalent attitudes. Honey and coconut oil were indicated as being healthy because they are “natural”. It is interesting to compare equivalent foods, such as coconut oil x soybean oil, processed turkey breast x ham, and gelatin x all other ultra-processed foods. For all stereotyped healthy foods, the professionals advocated some health properties that they could not identify in other foods that are very similar. For example, many professionals stated that processed turkey breast and gelatin have good nutritional compositions, but they did not do so for ham or fruit in syrup.

The users showed a similar response pattern, using the same categories and subcategories for professionals (see Appendix A for the detailed responses). However, the users mentioned fewer negative aspects of healthy stereotyped foods.

## 4. Discussion

This study aimed to assess food literacy and health professionals’ perceptions of the risks and benefits associated with food. More than half of the participants (58.93%) reported that they had never taken part in a course on food and nutrition, showing a gap in the education of professionals on this topic. Such courses and training are of utmost importance as they improve food literacy, especially in primary health care [30]. Primary health care is the gateway for users of the Brazilian health system and must meet the corresponding requirements, focusing on solving the problems of the population through the promotion, protection, diagnosis, and maintenance of health [15]. In this sense, Pinto (2021) highlighted that users of the Brazilian health system believe that nutrition is a prevention factor for different diseases, highlighting the importance of this issue in the approach taken by health professionals [31].

The mean food literacy score reported by the professionals (mean = 34.5) was lower than might be expected. In a previous study involving the general population, we observed a similar mean score (mean = 33.2) [25]. Since public health workers are in a position to provide education on healthy eating habits, a higher value might be expected. This shows the importance of educating this target group with regard to food and nutrition, especially in light of the Food Guide for the Brazilian Population, a document with which the professionals in this study had little familiarity. The lack of knowledge of this document among health professionals was also noted by Reis and Jaime (2019) [20]. This food guide is considered a reference in the Brazilian health system for the presentation of information and recommendations on nutrition, with a focus on health promotion. When used by primary health care workers, it ensures wide dissemination of its content and that the population understands it.

Regarding food risk and benefit perceptions, the professionals generally reported perceptions consistent with the degree of food processing. People generally judge vegetables, fruits, and white meats as good, pure, and healthier foods. In contrast, foods high in sugar and fat—usually industrially produced foods—are called “impure” and “unhealthy” [32]. According to Carels et al. (2006) [33], people use nutrients, fat, and sugar to describe how healthy a diet is. Nevertheless, some processed foods, such as gelatin, cereal bars, cream crackers, processed turkey breast, and whole-grain bread, were considered healthy or neutral. These healthy stereotyped foods demonstrate a bias that is present in the message these products convey. These foods are already perceived as healthy by a sample of Brazilian consumers [26]. Advertising can influence food behaviors on several levels. It contributes to the formation of a general knowledge about eating habits that is fragile and does not enable individuals to make healthy and independent choices [34]. This effect seems so strong that it even affects health professionals responsible for designing messages about health and nutrition.

Food literacy was a positive predictor of a better understanding of food health. Professionals with higher food literacy had less-biased perceptions of ultra-processed foods, meaning they knew that these foods may be unhealthy. This finding underscores the need to invest in professional development. Studies have reported that health professionals may have difficulty interpreting food and nutrition information [35,36]. The Brazilian National Food and Nutrition Policy defines the qualifications of professionals in line with the population’s health, food, and nutrition needs as essential. This represents a historical and strategic need to address the problems arising from the current Brazilian nutrition scenario [3]. Health professionals trained in nutritional issues can provide better patient care and thus improve the population’s health [37]. The training of health professionals should transcend the basics and aim to increase professionals’ risk perception and food literacy and educate them about food myths.

In the qualitative step, we aimed to scrutinize health professionals’ perceptions of foods, especially healthy stereotyped food. The perceptions of unhealthy foods were consistent with the Food Guide for the Brazilian Population [21]. Interestingly, health experts looked for positive attributes in healthy stereotyped foods, that is, ultra-processed foods with a similar composition to those perceived as unhealthy. For example, cream crackers were considered healthy because of their good composition. Light foods and processed turkey breast were reported to be low in calories or fat and have fewer preservatives. Gelatin was described as rich in calcium and collagen, while honey was highlighted as a food with antioxidants. These claims are consistent with common sense and do not justify the recommendation of these foods. In general, people tend to classify foods according to the dichotomy of “good” or “bad”, and it is well known that food choices are influenced by individual factors such as subjective aspects, knowledge about food and nutrition, and one’s beliefs about eating [38]. In addition, collective determinants such as economic, social, and cultural factors also play a significant role in these choices [39].

The users also had a biased perception of food. Of note is the exponential growth in the production and consumption of ultra-processed products. In contrast, a decline in the consumption of fresh or minimally processed foods and culinary preparations was observed [40]. The processes and ingredients used to produce ultra-processed foods are inexpensive, are associated with a strong brand, and a long shelf life. In addition, the convenience of these products (e.g., being ready-to-eat or nonperishable), their hyper-tastiness, and advertising provide ultra-processed foods with market advantages [40]. Aggressive marketing strategies use attractive and sophisticated food packaging to make food desirable [41], leading to a distorted perception that such food may have health benefits. Because these packages contain various information, purchasing decisions can become complex for consumers [42].

The practical implications of this work indicate the need to improve the food literacy of primary care health professionals. Since many professionals do not have a college degree, on-the-job education would be an appropriate strategy. Tramontt and Jaime (2020) [43], for example, have developed a workshop for primary healthcare professionals. They observed positive improvements in food knowledge and self-efficacy. Additionally, their study emphasizes the importance of a nutritionist being part of the team, guiding professionals and the population regarding food and nutrition. 

This study has some limitations. First, all professionals from a single Brazilian city were included in the study. Cities in the southeastern region of Brazil have better social indicators, on average, so the results may not reflect the reality of the entire country. Many health professionals (e.g., community health agents) do not require higher education to work in primary care. As most employees fit into this category (49.3%), this aspect is likely to have reduced the overall score for FL.

## 5. Conclusions

Health professionals in Brazilian primary care showed adequate FL, similar to the general population, with some difficulties regarding understanding and applying the concepts of the Food Guide for the Brazilian Population. Since the participants are public health workers who have the opportunity to influence eating habits, higher FL might be expected. FL was associated with some factors related to the perception of the risks and benefits of food and was a positive predictor of a better understanding of healthy eating. Professionals with higher FL had less-biased perceptions of ultra-processed foods. Some ultra-processed foods were assigned positive factors consistent with common sense rather than official recommendations. Users also indicated misperceptions of such foods. These findings highlight the need to improve FL among primary care professionals and invest in users’ education.

Educational processes for this target group must transcend the classroom. This group needs to improve its risk perception and communication and demystify its misperceptions and common sense regarding food.

## Figures and Tables

**Figure 1 nutrients-16-00302-f001:**
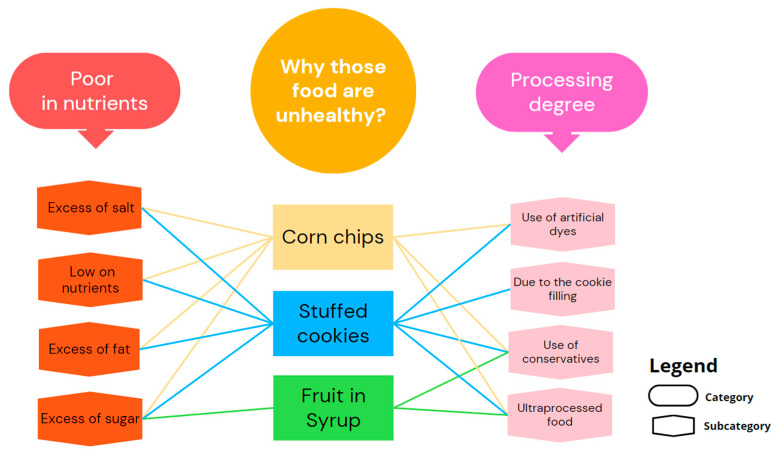
Categories and subcategories that emerged in the interviews when all participants (professionals and users) were asked about perceived unhealthy foods. Each line indicates whether a specific subcategory for a food has been cited, with lines of the same color corresponding to the same food.

**Figure 2 nutrients-16-00302-f002:**
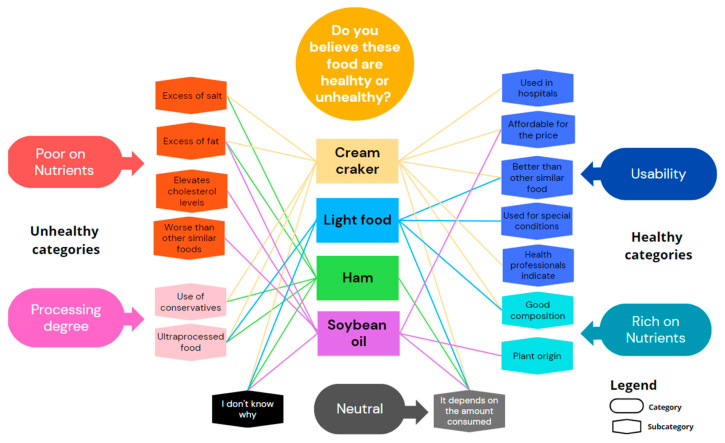
Categories and subcategories that emerged in the interviews when all participants (professionals and users) were asked about foods that are considered neither good nor bad for one’s health. Each line indicates whether a specific subcategory for a type of food has been cited, with lines of the same color corresponding to the same food.

**Figure 3 nutrients-16-00302-f003:**
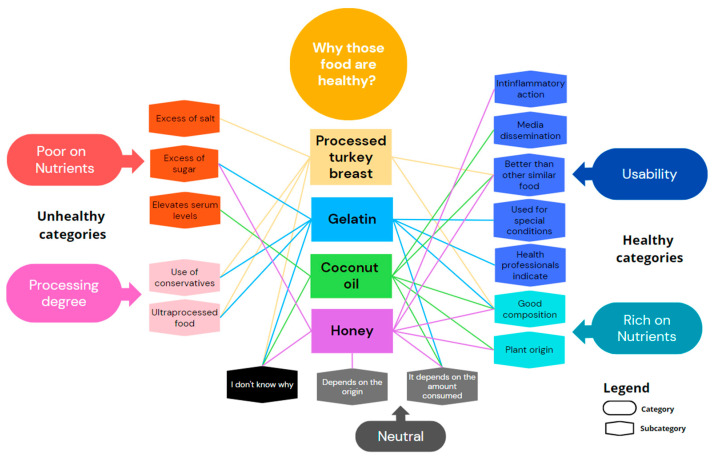
Categories and subcategories that emerged in the interviews when all participants (professionals and users) were asked about perceived healthy foods. Each line indicates whether a specific subcategory for the food has been cited, with lines of the same color corresponding to the same food.

**Table 1 nutrients-16-00302-t001:** Characteristics of primary health care professionals in Rio Claro, Brazil.

Variable	Categories	n	%
Gender	Women	242	86.43
	Men	38	13.57
Age range	20–30 years old	48	17.14
	31–59 years old	210	75.00
	≥60 years old	22	7.86
Education level	Complete primary	5	1.79
	Incomplete secondary	4	1.43
	Complete secondary	163	58.21
	Complete higher education	108	38.57
Function	Physician	13	4.64
	Nurse	18	6.43
	Nurse assistant	53	18.93
	Community health agent	138	49.29
	General service assistant	22	7.86
	Dentist	28	10.00
	Pharmacist	6	2.14
	Health manager	2	0.71
Have you taken part in a food and nutrition course?	Yes	115	41.07
	No	165	58.93

**Table 2 nutrients-16-00302-t002:** Mean, standard deviation, and factor loading of the SFLQ-Br indicators for primary health care professionals in Rio Claro, Brazil.

Indicators	Mean (SD)	Range	Factor Loading
1. When I have questions on healthy nutrition, I know where I can find information on this issue	3.21 (0.98)	0 to 4	0.427
2. In general, how well do you understand the following types of nutritional information? (Mean value)	3.48 (0.92)	0 to 5	0.413
2a. Nutrition information leaflets.	3.28 (1.52)	0 to 5	*
2b. Food label information.	3.29 (1.33)	0 to 5	*
2c. T.V. or radio program on nutrition.	3.31 (1.49)	0 to 5	*
2d. Oral recommendations regarding nutrition from professionals.	4.04 (1.22)	0 to 5	*
2e. Nutrition advice from family members or friends.	3.32 (1.39)	0 to 5	*
3. How familiar are you with Dietary Guidelines for the Brazilian Population?	2.63 (1.09)	1 to 5	0.620
4. I know the official Dietary Guidelines for the Brazilian Population recommendations about the consumption of in natura and minimally processed food.	2.36 (1.01)	1 to 4	0.768
5. I know the official Dietary Guidelines for the Brazilian Population recommendations about oil, fat, salt and sugar intake.	2.45 (1.06)	1 to 4	0.838
6. Think about a usual day: how easy or difficult is it for you to compose a balanced meal at home?	2.50 (0.91)	0 to 4	0.465
7. In the past, how often were you able to help your family members or a friend if they had questions concerning nutritional issues?	3.13 (0.95)	0 to 5	0.653
8. There is a lot of information available on healthy nutrition today. How well do you manage to choose the information relevant to you?	3.77 (0.94)	0 to 5	0.652
9. How easy is it for you to judge if media information on nutritional issues can be trusted?	2.65 (0.67)	1 to 4	0.644
10. Commercials often relate foods with health. How easy is it for you to judge if the presented associations are appropriate or not?	2.71 (0.69)	1 to 4	0.728
11. How easy is it for you to evaluate if a specific food is relevant for a healthy diet?	2.81 (0.61)	1 to 4	0.660
12. How easy is it for you to evaluate the longer-term impact of your dietary habits on your health?	2.86 (0.71)	1 to 4	0.513

* Included as mean value of all number 2 indicators; SD = standard deviation.

**Table 3 nutrients-16-00302-t003:** Means, standard deviation, and factor loadings of indicators of food risk and benefit perceptions of primary health care professionals in Rio Claro, Brazil.

Indicator	Mean ‡ (SD)	Factor Loading
Factor 1—Healthy Stereotyped food	-	-
Gelatin	1.14 (1.51)	0.397
Cereal bar	0.89 (1.36)	0.579
Cream crackers	0.07 (1.45)	0.632
Processed turkey breast	1.10 (1.58)	0.542
Whole-grain bread	0.98 (1.26)	0.578
Factor 2—Cafeteria	-	-
Stuffed cookies	−2.07 (1.07)	0.651
Instant noodles	−2.12 (1.19)	0.721
Soft drinks	−2.53 (1.00)	0.807
Ham	−0.68 (1.57)	0.647
Fruit in syrup	−1.04 (1.42)	0.559
Mozzarella cheese	0.04 (1.55)	0.533
Margarine	−1.70 (1.42)	0.659
Corn chips	−2.58 (0.86)	0.784
Factor 3—Typical Brazilian staple foods	-	-
Rice	1.04 (1.33)	0.387
Baked potatoes	1.65 (1.04)	0.456
Beans	2.24 (0.75)	0.261
Whole milk	0.33 (1.64)	0.542
Pasta	0.24 (1.41)	0.686
White bread	−0.45 (1.35)	0.847
French white bread	−0.30 (1.39)	0.830
Factor 4—Fad diet foods	-	-
Chicken breast	2.15 (0.85)	0.797
“Minas frescal” cheese †	1.82 (0.90)	0.646
Hearts of palm	1.70 (1.07)	0.306
Tomato	2.34 (0.74)	0.554
Egg	2.39 (0.62)	0.677
Factor 5—Ultra-processed food	-	-
Ready seasonings (salt and fat added)	−2.44 (0.87)	0.736
Processed juice (sugar added)	−2.00 (1.19)	0.810
Powdered juice	−2.53 (0.78)	0.867
Instant dry soup	−2.25 (1.01)	0.808
Factor 6—Foods for special dietary uses	-	-
Sweetener	−0.95 (1.64)	0.754
Sugar-free food	0.42 (1.59)	0.868
Light food *	0.58 (1.58)	0.875
Factor 7—Oils and fat	-	-
Canola oil	0.85 (1.64)	0.579
Soy oil	−0.49 (1.67)	0.691
Foods without a defined factor	-	-
Avocados	2.33 (0.73)	-
Refined sugar	−2.06 (1.23)	-
Lettuce	2.56 (0.84)	-
Olive oil	2.29 (0.83)	-
Bananas	2.59 (0.63)	-
Pork fat	0.67 (1.73)	-
Coffee	0.67 (1.48)	-
Red meat	1.42 (1.36)	-
Canned peas	−0.75 (1.53)	-
Butter	0.78 (1.57)	-
Honey	2.10 (0.84)	-
Canned corn	−0.72 (1.46)	-
Coconut oil	1.92 (1.01)	-
Refined salt	−1.32 (1.54)	-
Pink salt	0.75 (1.58)	-
Canned vegetables	−0.64 (1.77)	-

* In Brazil, all food with low or reduced levels of fat, sodium, or energy are called light foods; † traditional Brazilian cheese, which is white-colored and low in fat; ‡ range: −3.0 to 3.0.

## Data Availability

Data is contained within the article (and Appendix A).

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
