# Peer review of "Do Primary Health Professionals in Brazil Have a Misperception about Food? The Role of Food Literacy as a Positive Predictor"

_nutrients, 2024, doi:10.3390/nu16020302_

Round 1

Reviewer 1 Report

Comments and Suggestions for Authors

In confirmatory factor analysis of foods perception, 16 foods were not included in any defined factor, including healthy food like olive oil, lettuce and so on, and unhealthy foods such as refined sugar and pork fat, which relationship with food literacy remained unclear.  In fact, except of factor analysis, some foods could be categorized as healthy or unhealthy foods generally according to their energy density and nutrient density, referring to dietary guideline. Then the relations of healthy and unhealthy foods perceptions with food literacy could be analyzed in health professionals.

Author Response

In confirmatory factor analysis of foods perception, 16 foods were not included in any defined factor, including healthy food like olive oil, lettuce and so on, and unhealthy foods such as refined sugar and pork fat, which relationship with food literacy remained unclear.  In fact, except of factor analysis, some foods could be categorized as healthy or unhealthy foods generally according to their energy density and nutrient density, referring to dietary guideline. Then the relations of healthy and unhealthy foods perceptions with food literacy could be analyzed in health professionals.

Response: Dear reviewer, we appreciate your suggestion. When selecting foods for the qualitative step, we used the order of magnitude (mean) and not a food from a specific group. We understood your suggestions, but we would like to do the analysis from the 'perception' point of view. It is curious that the professionals could not associate some foods with certain patterns. This should be investigated further. The problem with "forcing" a group based only on food characteristics is the low reliability of the factor and therefore poor statistical analysis. If we force a group based on nutritional aspects, the stereotyped food factor would not exist.

Reviewer 2 Report

Comments and Suggestions for Authors

I really liked this.  I have included some comments in the attached PDF and add some broad brush stroke comments below.

I think the use of the term bias is misleading and not the correct term.

I would recommend the title be amended see comments in PDF

Are you looking at the issue of nutrition knowledge and not the wider issue of food literacy.  As I noted in the PDF I can be food literate for my own issues but not capable of passing on the knowledge to others and the transfer of knowledge into skills. My big question for you is were you exploring issues of nutrition and dietary knowledge and not the wider issues of food literacy and how skilled the primary care professionals were in doing this?  In many countries primary care professionals have to do a teaching or learning certificate to enable this. 

You need to expand your discussion on food literacy and offer a more curtail stance (line 52+).

Issues of public health or population approaches and clinical encounters are not the same. Lines 75 + and  82+ seems confused over these issues, of course individual or family clinical work can be part of a wider public health approach, but population approaches are more generic. 

I have added some comments requesting information on nutrition education what primary care professionals receive. perhaps it is best left to dieticians and nutritionists as they have the knowledge?? In some instances the correct approach might be to refer on to the nutrition or food literacy experts. 

The three figures are diagramatic representations of the findings but contain no detail, we as readers need some information on the results and how these were sued to generate the figures. Also I am not clear what  or what the 16 'users' contribute to the overall findings. Nor am IN clear what they were asked, the veracity or usefulness of the food literacy advice? You say 

Based on the quantitative step, some foods were selected for interviews to explore 266 perceptions of the risks and benefits of these foods and their effects on health. These were 267 corn chips, stuffed cookies, and fruit in syrup (with negative mean scores, suggesting they 268 are harmful to health); cream crackers, ham, soy oil, and light foods (with positive or neg-269 ative mean scores near zero, suggesting they are neither good nor bad for health); and 270 processed turkey breast, gelatin, coconut oil, and honey (with positive mean scores, suggesting they are good for health). 

But how does this relate to the encounters with primary care professionals and what were the nature of these encounters?

Comments on the Quality of English Language

There are many superfluous sentences and descriptions eg' The face-to-face interview is the most commonly used strategy in fieldwork'; is it and where is the citation to justify this?

Author Response

I really liked this.  I have included some comments in the attached PDF and add some broad brush stroke comments below.

Response: Dear reviewer, I would like to thank you for your careful review and suggestions. We have made all the corrections you pointed out in the pdf file. 

I think the use of the term bias is misleading and not the correct term.

Response: We agree with the reviewer. We changed bias for misperception. 

You need to expand your discussion on food literacy and offer a more curtail stance (line 52+).

Response: We included more information about food communication in line 70.

About the hability to share information, we included three categories of food literacy: functional, interactive and critical. 

Issues of public health or population approaches and clinical encounters are not the same. Lines 75 + and  82+ seems confused over these issues, of course individual or family clinical work can be part of a wider public health approach, but population approaches are more generic. 

Response: We agree with the reviewer. We improved this section to clear provide how the primary health care workers work. 

I have added some comments requesting information on nutrition education what primary care professionals receive. perhaps it is best left to dieticians and nutritionists as they have the knowledge?? In some instances the correct approach might be to refer on to the nutrition or food literacy experts. 

Response: Thank you for this comment. We fully agree with you, however, the nutritionist (or dietician) is not a common professional in primary care in Brazil. Therefore, we have discussed the nutritional competence of all other health professionals. We have mentioned in the practical implications the importance of nutritionists in primary care.

The three figures are diagramatic representations of the findings but contain no detail, we as readers need some information on the results and how these were sued to generate the figures. Also I am not clear what  or what the 16 'users' contribute to the overall findings. Nor am IN clear what they were asked, the veracity or usefulness of the food literacy advice? You say

Response: We agree with the reviewer. However, it was really difficult to find a practical way to show these results. We have attached an additional file (supplementary file) to the article with the details of all the qualitative analysis. In this file you can see how many professionals and users we have in each category or subcategory. We agree with you about the users. However, our intention was to show that the perception of users is not far from the perception of professionals. This is the core problem: how can a professional educate a user without the relevant knowledge and skills? We would like to keep the users' results for comparison.

Based on the quantitative step, some foods were selected for interviews to explore 266 perceptions of the risks and benefits of these foods and their effects on health. These were 267 corn chips, stuffed cookies, and fruit in syrup (with negative mean scores, suggesting they 268 are harmful to health); cream crackers, ham, soy oil, and light foods (with positive or neg-269 ative mean scores near zero, suggesting they are neither good nor bad for health); and 270 processed turkey breast, gelatin, coconut oil, and honey (with positive mean scores, suggesting they are good for health). 

But how does this relate to the encounters with primary care professionals and what were the nature of these encounters?

Response: Thanks for the remark. We used the foods to base que interview (please see section 2.2.2 of methods). The interviews were structured and characterized by a combination of questions requiring the interviewees to express themselves on the topic, with information derived from their reflection on reality and their experiences. Both target groups were asked questions about why they think certain foods are healthy or unhealthy, e.g. “Why do you think “corn chips” are unhealthy? The foods were strategically selected after analyzing the quantitative data, i.e., for the unhealthy food question, foods with a rating of -1.0 or less were selected; for the neutral food question, foods with a rating between -0.9 and 0.9 were selected; and for the healthy food question, four foods were selected from the stereotyped healthy food.

Reviewer 3 Report

Comments and Suggestions for Authors

Diogo Thimoteo da Cunha et al. submitted to Nutrients an article focusing to the role of food literacy as a predictor for primary healthcare workers to biased perception towards food.

This manuscript is well structured and makes for useful reading for experts in the field.

Below are my suggestions for improvement.

- Please specify in detail, in the discussions, the practical methods through which the university teaching system in your Country could provide for feasible changes to improve the outcomes on a topic such as the one described (community of practice laboratories, supplementary courses, compulsory curricular teachings, etc.);

- you need to better describe the limits of this study, currently extended to two lines only;

- it is not clear why some citations are indicated in parentheses and others in square brackets.

Comments on the Quality of English Language

Minor editing of English language required

Author Response

Reviewer:

Diogo Thimoteo da Cunha et al. submitted to Nutrients an article focusing to the role of food literacy as a predictor for primary healthcare workers to biased perception towards food.

This manuscript is well structured and makes for useful reading for experts in the field.

Below are my suggestions for improvement.

Response: Thank you for your comments and suggestions.

- Please specify in detail, in the discussions, the practical methods through which the university teaching system in your Country could provide for feasible changes to improve the outcomes on a topic such as the one described (community of practice laboratories, supplementary courses, compulsory curricular teachings, etc.);

Response: As mentioned in the text, we do not believe that this is only a problem of high education. The team needs a nutritionist to help them and train the team to multiply the knowledge. We included some sentences to better elucidate this ideia, as follows:

"The practical implications of this work indicate the need to improve the food literacy of primary care health professionals. Since many professionals do not have a college degree, on-the-job education would be an interesting strategy. Tramontt and Jaime (2020) [42], for example, have developed a workshop for primary healthcare professionals. They observed positive improvements in food knowledge and self-efficacy. Additionally, it emphasizes the importance of the nutritionist being part of the team, guiding professionals and the Population regarding food and nutrition."

- you need to better describe the limits of this study, currently extended to two lines only;

Response: We agree with the reviewer. We have included more information on limitation section, as follows:

"This study has some limitations. First, all professionals from a single Brazilian city were included in the study. Cities in the southeastern region of Brazil have better social indicators on average, so the results may not reflect the reality of the entire country. Many health professionals (e.g. community health agents) do not require higher education to work in primary care. As the majority of employees are in this category (49.3%), this is likely to have reduced the overall score for FL."

- it is not clear why some citations are indicated in parentheses and others in square brackets

Response: Thanks for the remark. We corrected the references using Mendeley. 

Round 2

Reviewer 2 Report

Comments and Suggestions for Authors

This is a resubmission which address the main issues raises in my first review, there remains. couple of minor issues which are included in the attached PDF. These in summary are 

No need to use capital P for Population. Just use population or population's.

The following should be changed from 

In nutrition, primary healthcare professionals face the challenge of providing food 91 and nutrition knowledge to the individuals 

To 

In nutrition, primary healthcare professionals face the challenge of providing food  and nutrition knowledge to the individual

Or to

In nutrition, primary healthcare professionals face the challenge of providing food and nutrition knowledge to individuals  

or to

In primary healthcare settings professionals face the challenge of providing food and nutrition knowledge to the individual 

or to

In primary healthcare settings professionals face the challenge of providing food and nutrition knowledge to individuals

This section requires a slight rewrite 

The mean food literacy score reported by the professionals (mean = 34.5) was consid-340 ered adequate (66%). In a previous study with the general Population, we observed a sim-341 ilar mean score (mean = 33.2) [26]. Since they are public health workers and influence the 342 education of healthy eating habits, a higher value was expected. This shows the im-343 portance of educating this target group on food and nutrition, especially in light of the 344 Food Guide for the Brazilian Population, a document with which the professionals in this 345 study had little familiarity. The lack of knowledge of this document by health profession-346 als was also noted by Reis and Jaime (2019) [21]. 

To something like 

The mean food literacy score reported by the professionals (mean = 34.5) was lower than might be expected (66%). In a previous study with the general population, we observed a similar mean score (mean = 33.2) [26]. Since public health workers are in a position provide  education on healthy eating habits, a higher value might be expected. This shows the importance of educating this target group on food and nutrition, especially in light of the 344 Food Guide for the Brazilian Population, a document with which the professionals in this study had little familiarity. The lack of knowledge of this document by health professionals was also noted by Reis and Jaime (2019) [21]. 

Substitute appropriate for interesting 

Since many professionals do not have a college degree, on-the-job education would be an interesting strategy. 

Is the following correct, 49.3% is not a majority?

As the majority of employees are in this category (49.3%), this is likely to have reduced the overall score for FL. 

Comments on the Quality of English Language

Now fine, see comments on capital P in populations and some rewording of sentences 

Author Response

Dear reviewer,

Many thanks for your suggestions and review.

This is a resubmission which address the main issues raises in my first review, there remains. couple of minor issues which are included in the attached PDF. These in summary are 

1) No need to use capital P for Population. Just use population or population's.

We agree with the reviewer. We changed all words. 

2) The following should be changed from 

In nutrition, primary healthcare professionals face the challenge of providing food 91 and nutrition knowledge to the individuals 

To 

In nutrition, primary healthcare professionals face the challenge of providing food  and nutrition knowledge to the individual

Response: We agree with the reviewer. Done it as suggested.

3) This section requires a slight rewrite 

The mean food literacy score reported by the professionals (mean = 34.5) was considered adequate (66%). In a previous study with the general Population, we observed a sim-341 ilar mean score (mean = 33.2) [26]. Since they are public health workers and influence the 342 education of healthy eating habits, a higher value was expected. This shows the im-343 portance of educating this target group on food and nutrition, especially in light of the 344 Food Guide for the Brazilian Population, a document with which the professionals in this 345 study had little familiarity. The lack of knowledge of this document by health profession-346 als was also noted by Reis and Jaime (2019) [21]. 

To something like :

The mean food literacy score reported by the professionals (mean = 34.5) was lower than might be expected (66%). In a previous study with the general population, we observed a similar mean score (mean = 33.2) [26]. Since public health workers are in a position provide  education on healthy eating habits, a higher value might be expected. This shows the importance of educating this target group on food and nutrition, especially in light of the Food Guide for the Brazilian Population, a document with which the professionals in this study had little familiarity. The lack of knowledge of this document by health professionals was also noted by Reis and Jaime (2019) [21]. 

Response: We agree with the reviewer. We changed the sentence as suggested.

4) Substitute appropriate for interesting 

Since many professionals do not have a college degree, on-the-job education would be an interesting strategy. 

Response: Done it as suggested.

5) Is the following correct, 49.3% is not a majority?

As the majority of employees are in this category (49.3%), this is likely to have reduced the overall score for FL. 

Response: We changed "majority" for "most"